# Rebuilding a Cluster While Protecting Knowledge within Low-Medium-Tech Supplier SMEs: A Spanish and French Comparison

**Martine Gadille** [1],* and **Juan Ramón Gallego-Bono** [2],*

1 Institute of Labor Economic and Industrial Sociology, Aix Marseille University CNRS LEST, 35 Avenue Jules Ferry, 13626 Aix-en-Provence, France
2 Department of Applied Economics, University of Valencia, Avda Tarongers s/n, 46022 Valencia, Spain
* Correspondence: martine.gadille@univ-amu.fr (M.G.); Juan.R.Gallego@uv.es (J.R.G.-B.)

**Abstract:** Most of SMEs are engaged in open innovation practices, but they do not benefit from open innovation or from patenting in the same way as larger firms do. At the same time SMEs, as territorialized suppliers, play a crucial role within evolving regional specialization. In this context the purpose of our study is to examine how low and medium technology supplier SMEs learn and organize themselves at a territorial level to address the challenge of IP protection in an open innovation paradigm. We used a qualitative method with a longitudinal multi-case study involving 27 companies with a historical lance to compare the territorial dynamics of knowledge protection within clustered supplier SMEs in two European regions. The results show they protect their knowledge by learning how to design, in a direct relationship with clients, customized complex technological products to develop a new organizational matrix of multidisciplinary knowledge that reveals itself difficult to imitate within the clusters. They also cope with other supplier firms across sectors even if they show societal path dependencies in the way to build cooperation. This dynamic has given birth to changing structural relationships among regionally clustered SMEs and between them and large firms.

**Keywords:** open innovation; intellectual property; low-medium tech suppliers SMEs; regional clusters; cooperation; organizational matrix; regional specialization; societal path dependency

## 1. Introduction

There is a lively debate in the literature on the differences in IP (intellectual property) protection mechanisms between large companies and SMEs (small and medium-sized enterprises) and on how they profit from them in an open innovation paradigm. The former still use offensive and expensive patent-lodging strategies [1–5]. They also adopt advantageous open innovation strategies [6,7]. Meanwhile, the latter show difficulties to profit from this mixed strategy [8–11]. As high-tech SMEs are often embedded within clusters, other authors have focused on the role of clusters in driving innovation while protecting IP within systems that are more open to knowledge sharing [12,13]. However, few of these results have been obtained with non-high-tech SMEs and, moreover, those embedded within clusters [14–16]. Hence, in a context of evolving clustering processes within specialized regions, our main question is how do LMT (low and medium technology) supplier SMEs learn and organize themselves at a territorial level in a new way to address the challenge of IP protection in an open innovation paradigm? Before answering this question, a few points should be clarified. In contrast to a minority of SMEs in high-tech sectors that undertake intensive use of science-based knowledge and innovate essentially through R&D, low- and medium-tech SMEs rely less on formal R&D and more on design-based innovations, as well as on their knowledge of product interdependencies and a deep understanding of customer needs [17]. Some authors also show that the

development of successful innovation collaborations in the low- and medium-tech sector requires SME managers to start by creating an internal culture which favors innovation, learning and openness to the external environment [18]. Consequently, by comparing how these companies in the two selected regions are organized in order to respond to the challenges of managing their intellectual property, our article aims to fill a gap in the innovation mode of low- and medium-tech SMEs by examining the territorial and societal contexts of clusters. However, the ability to undertake significant R&D efforts also depends on firm size. Hence, our supplier sample includes not only SMEs but also some large firms in order to capture the importance of dynamic interactions between large firms and SMEs in the context of clusters.

To answer this question, we will rely on two main kinds of results in the literature. In a first approach, the authors focused on IP protection practices such as secrecy and lead time in SMEs in general [19,20], assuming that the number of innovative SMEs concerned by this issue is growing. There is a tension between cooperation and secrecy to innovate continuously [21]. This makes 'lead time' very demanding and hard to achieve [20]. Confirming and comforting the results of this initial literature, Van de Vrande et al. state that the ability of SMEs to protect their IP while cooperating implies overcoming managerial, organizational and cultural barriers and that academia should deeply understand how clustering SMEs manages this balance [10]. The paradigm of open innovation would not imply the same learning and risk for SMEs and large companies [22–24]. However, this literature does not deal with clustered SMEs within a framework of agglomeration economies integrating learning in cities and their microeconomic foundations [16,25,26]. Within this second field of literature, a recent trend is of great interest to understand how LMT supplier SMEs could protect their IP within clusters. In line with the previous work of Doeringer and Terkla, it highlights the capacity of mobile and networked entrepreneurial workers to achieve new technological combinations crossing sectoral boundaries [27–29]. These entrepreneurial workers incorporate something new into their current activity through their technical creativity or through a new use for their existing expertise in the same territory [30]. This creativity would have important analytical consequences on IP protection strategies. It suggests that SMEs supplying machinery and technological services specialize in territories, but at the interface of different sectors, thus accessing different markets. In coherence with this view, Breznitz and Taylor relate the success of innovations with rich multiple, locally centered social networks where the nature of cohesive and social structures underpin cooperation [31]. This literature strengthens the evidence to embed the strategy, organization and culture of the IP protection of supplier SMEs within specific territorial cultures and identities [24,32,33].

The research methodology follows two essential parts. On the one hand, from the study of the theoretical and empirical literature, three basic propositions were derived (which operate approximately at the micro, meso and macro levels), propositions which are to be contrasted precisely with our own empirical evidence. On the other hand, and in coherence with the qualitative concern regarding the dynamic aspects of the article, the empirical research itself is essentially based on in-depth interviews with a group of companies and experts from two clusters in two regions. These are the Comunitat Valenciana (Spain) and Provence (France). The fact that a large number of companies (27 in total) were interviewed at various points in time makes this a truly longitudinal, multi-case study.

Based on this methodology, we argue to a large extent, that the strategy of low- and medium-tech SMEs to reconcile innovation protection and the use of knowledge and external cooperation stems from the threat posed by the strong offensive IP protection strategies of large competing firms. However, thanks to the integrative and transversal competences developed by SMEs and their ability for product diversification, their formerly conflictive relationship with large competitors now has changed. They have developed more cooperative relationship through a better balance of power. Thus, we are witnessing a rebuilding of clusters on the double basis of a shift in the relations between large supplier companies and

SMEs, on the one hand, and an intensification of intersectoral relations between existing clusters within the territory and the region, on the other. This process of interaction and an increasing intra- and inter-cluster intersectoral integration makes it possible to deal with lock-in or structural blockage risks associated with the sectoral specialization of clusters, because it endows them with greater technological and institutional-relational complexity.

The main contribution of the article consists in the fact that this empirical research was carried out using an approach that combines the existing literature on the intellectual property protection of large firms and SMEs in an open innovation paradigm by incorporating the problems that define this type of process in the territorial context of clusters. This approach makes it possible to integrate the triple micro (individual business strategies), meso (local business networks and networks with other actors, intra- and inter-sectoral) and macro (national and regional economy as a whole) dimensions of the change process. Thus, it seems clear that the comparison between the two dynamics of clusters makes it possible to understand the role played by the institutional and territorial context in which the SMEs innovation and intellectual property protection strategies are deployed. A context that not only helps to apprehend the process of change but also the differences described in the two regions and countries considered. At the same time, it makes two significant contributions. Firstly, on the cluster literature by showing the importance of IP protection in an open innovation context and secondly for understanding the role of low- and medium-tech SMEs in the transformation of clusters, an angle of attack that has seldom been explored so far.

The paper proceeds as follows: an overview, the description of the methodology and data set, a presentation of the socio-economic background of the two compared regional economies and the comparative results between the two regional clusters, a discussion and a conclusion.

## 2. Literature Review

In the first section, we analyze the literature, pointing out that the patenting system reveals asymmetries linked to size effects and the technological evolution of the division of work between large firms and SMEs within a strengthened IP regime. Then, in the second section, we present the main literature focused on SME strategies to adapt to these asymmetries and their convergence toward the open innovation paradigm. The third section addresses the consequences that a mix between a stronger IP regime and an open innovation paradigm has on an SME's IP strategies and innovation at the cluster level.

### 2.1. Asymmetries within IP Regimes between Large Multinational Firms and SMEs

Evolving IP rights can have differentiated effects on the way businesses structure their markets [34–37]. They also shape the kinds of skills and knowledge that pay off [38,39]. In the case of patents, several authors have pointed out the growing dysfunctionality in the US patents system during the 1990s: simple ideas that prove commercially profitable are patented when they have already been implemented by competitors [1,5,40]. At the world level, Lerner finds that the impact of institutional changes in favor of patenting on applications to file a patent by residents as opposed to foreign firms is negative, which we can see as a 'puzzling result' [41] (p. 347). This converges with the fact that when sectoral effects are controlled for, the propensity to patent in different countries increases with the size of firms [42–47]. Nevertheless, all sectors have been confronted with this strengthening IP regime, where patents are the results of purposeful, routine corporate R&D [48–51]. One reason is that large multinational firms have developed new strategic management of their IPR (intellectual property rights) portfolios to improve their reputation, to obtain leverage during negotiations, and to motivate their R&D employees [52,53]. Meanwhile, those engaged in the past to protect knowledge from imitation and to block product development from their competitors in an offensive way also continue to do so [37,50,51,54,55]. As a consequence, these strategies cannot fail to have differentiated effects on the positions of most SMEs as they are involved in the strong tendency of the whole economy of industrial

countries toward a technological regime based on complex technologies [50,51,56–62]. In a strengthening IP regime, the literature acknowledges that when SMEs use formal rights to protect their IP, they do not use them in the same way as larger firms. Firstly, for reasons of fixed costs, it can be too costly for SMEs to patent in the same way as large firms [53]. These costs can also be different according to the national legal context, for example when they do not benefit from legal adaptations, as used to be the case in Europe with fee schedule payment, compared to Japan and USA, where for many years these entities paid only 50% of the fees [61–65]. Secondly, most manufacturing SMEs cannot afford costly and time-consuming internal or external expert services to defend against patent infringement [19,50]. However, the failure to patent also remains risky, even in the case of LMT supplier SMEs. They may worry about losses of IP with imitators or about their reputation on similar markets if they do not protect themselves defensively [50,66]. This converges with the fact that, while most patenting SMEs show a greater proportion of patents leading to licenses and a smaller proportion of unused patents compared to large firms, this is in view of convincing investors or generating new revenue [67]. However, in doing so they give information to competitors through patent databases, which can be detrimental to the profitability of their intellectual assets [68]. Consequently, they resort essentially to informal time- and cost-saving practices and methods embedded in everyday routines, such as secrecy and lead time [19,21,62]. Complex design is also cited by some studies as an informal method to protect IP [55,69].

The main interest of formal rights (patenting, contracts, trademarks, etc.), perceived by manufacturing SME owners as time- and cost-consuming, depends on their capacity to deter infringement rather than to their potential to receive compensation through costly litigation over the misuse or loss of IP [19,66,70]. To a lesser extent, this argument is relevant when SMEs want to insure their reputation [33,66,71]. Secrecy or lead time may also facilitate innovation appropriation in terms of making the boundary between the firm and outside organizations less permeable in contexts where it is hard to protect invention [55,62,69,72,73]. To a lesser extent, they also use IP clauses within contracts, but their utility is reduced through the difficulty of forging trusty relationships with other important actors by means of stable and long-term relationships, a general characteristic of SME owners [19,62,71,74].

This asymmetry is still apparent within the open innovation paradigm that emerged during the 2000s. Large firms in technology-intensive industries, measuring the risk of strong IP protection management within a complex system technological regime, engaged in inside-out open innovation practices to value internal knowledge through activities such as venturing and licensing with others, including technology SMEs [75,76]. They also engaged in outside-in activities, such as acquiring new knowledge from outside the firm, involving small firms specialized in R&D that are forced to rely on the commercialization of external technology due to their limited resources [8,9,48,77–79]. However, the open innovation model does not benefit large firms and SMEs in the same way. When they collaborate with large firms, SMEs can be confronted with appropriation concerns [50,80,81]. Thus, the external sourcing of ideas or knowledge assets does not necessarily improve an SME's performance relative to larger firms [82]. It may be in their best interests to pursue a closed innovation strategy [24,82,83]. In the qualitative work of Eppinger and Vladova, it is explained that SMEs do not have the same finances, skills and specialized services, including specialized high-tech activities [22]. Even in high-tech industries, SMEs are not able to exploit external knowledge bases and to use outsourcing opportunities in the same way as large firms [22,62,84]. Brem, Nylund, and Hitchen confirmed at the national level of Spain that SMEs do not benefit from open innovation or from patenting in the same way as larger firms [11].

Hence, our first proposition is the following: even in an open innovation paradigm, SMEs show more weaknesses than large firms do to protect their knowledge and innovations through strong and formal IP rights.

### 2.2. SME Specific Organization and Strategic Adaptations toward Open Innovation

Several authors agree that SMEs have taken an increasingly prominent role in open innovation-related activities [10,14,84–88]. Hence, SMEs cannot share the same open innovation model as larger firms [23,60,71,89].

Firstly, the importance of a user-centered relationship in the innovation process with customer involvement to collect ideas is strongly confirmed for SMEs engaged in open innovation practices [10,23,24,69]. This assumption is in line with the work of Lundvall [90] and Von Hippel [91]. They also cooperate with a few other parties in order to develop new products or services based on their own in-house development knowledge [92] but need to decide on what they are not going to share [69]. Nevertheless, the implementation of this kind of decision does not appear to be an easy task due to the undesired transfer of information that can occur in the continuous relation with other actors. This could explain the selectivity of SME owners with regard to the nature and number of partners [20,21,24,33,93].

Secondly, on the exploitation side, SMEs consider outside collaboration and external sources as a means of access to marketing and sales channels and use them at the latest stages of innovation for the commercialization phase of their technology [33,71,72,92,94].

Hence, very few SMEs practice outward and inward IP licensing, venturing activities and external participations. Instead, they select practices like customer involvement and external networking, which are informal unstructured practices that do not necessarily require substantial investment [10,32,62,69]. This finding is in line with earlier works studying the practices that SMEs use to protect their intellectual assets while collaborating with a few partners [21].

This persistence of the inbound open innovation of SMEs is tightly combined with tacit knowledge sourcing in the proximity of clients and a few cooperation relationships with other parties [89]. This result is still coherent with the work of Pavitt and Bell and Pavitt who identify specialized SMEs as a category toward technological change analysis [95,96]. In these specialized supplier SMEs, product technologies develop incrementally from their earlier operating experience and improvements in components, machinery and sub-systems based on a design inspired by their daily contact with their clients' experience [96,97]. Bell and Pavitt [96] have emphasized the monitoring of advanced users' needs and integrating new technology into products as a main IP strategy. These results were confirmed more recently in the metal and machinery sector by Hall et al. [46], who highlight a predicted patenting propensity as a function of size, even if supplier SMEs have recently shown a tendency to patent. A result in line with Hall et al. [46] is that some authors also argue that process innovation through user/producer relations can be an alternative to product innovation, which is both easier to patent and to imitate, because reverse engineering of process innovation is more difficult to undertake for competitors, which makes secrecy more efficient [55,66,97]. However, this risk is also relevant to clients because of their capacity to divulge to other suppliers or because of a partial return on information about product innovation contributions to their value creation [96,98]. In the open innovation paradigm, SMEs are still supposed to have a tendency to fear collaborating on innovation with other partners in order to protect their IP [99]. They mainly fear intentional or accidental leakage [100,101]. Relying on qualitative studies, several authors suggest that SMEs lack an explicit IP strategy and coherent IP management [22]. They also need to collaborate because of a lack of multidisciplinary competencies [24,102–104].

It is also argued in a more evolutionary view that SME learning capabilities alongside sales or service efforts and lead time can be more effective than both patents and secrecy [43,55,62,66]. This view leads us to better consider how companies should reinforce their external collaboration and internal knowledge management processes to generate innovations [105]. This learning perspective suggests some entrepreneurial critical self-reflection and reflexivity about IP management and is in line with the framework of entrepreneurial learning relying on Mezirow's transformative learning theory. As defined by this author transformative learning is "the process by which we transform problematic

frames of reference (mindsets, habits of mind, meaning perspectives)—sets of assumption and expectation—to make them more inclusive, discriminating, open, reflective and emotionally able to change" [106] (p. 92). Entrepreneurial learning highlights the role of entrepreneurs' capacities to learn from the experience of failure and turn it into learning outcomes and more sophisticated entrepreneurial mental models [107,108]. All of these features correspond to a type of firm close to an operating adhocracy, building expert and embodied knowledge through experimentation and interactive problem solving [109,110]. In such SMEs, the effect of external collaboration on innovation is mediated by organizational learning [105]. This literature suggests that the knowledge base of supplier SMEs has evolved so that they can no longer have strong concentration on their core business and a specialized knowledge base, at least as it was analyzed in the 1990s [111,112]. Insofar as supplier firms are cross-sectoral in their technological offer, they require a basis of multidisciplinary competence that can be difficult to build at the firm's internal level [76,97,102]. Learning capabilities are all the more at stake in LMT supplier SMEs due to the differences between industries and high-tech or low-tech sectors, which seem to decrease in an open innovation paradigm [14,32,69,97,113].

These arguments lead us to our second proposition, which is the following: in an open innovation paradigm, LMT supplier SMEs grasp ideas mainly from close relationships with clients' needs from different industries to develop internal multidisciplinary knowledge on the basis of which they can balance cooperation to innovate and IP management, involving formal and informal means of protection.

However, other literature highlights that to develop a more fine-grained understanding of how these SMEs cooperate and appropriate innovation, they must also be studied at a regional and clustered level, where the relational density of localized networks could influence the form of organized action and extra-territorial linkages [14,31,89,114–116].

### 2.3. SME Knowledge Mobility and Appropriability within Regional Cluster Dynamics

There is still little work on how LMT supplier SMEs protect their IP in cluster dynamics, despite the amount of work on clusters. Thus, in coherence with what has been said above, we include the literature focused on IP management and cooperation dynamics of high-tech SMEs within an agglomeration context. In this literature, industry–university relations, adequate national or international IPR, and venture capital (enabled through expert uses of these IPR), have been identified as pre-existing or co-evolving conditions of North American and European high-tech clusters [12,117–123]. These scholars have taught us that the search for profitable and efficient IP management that is both individual and collective at the local level, is an important vector of high-tech cluster dynamics within a changing national regulatory framework [12,124–127]. In the case of information technology clusters, results also insist on the causal relationship between the extent to and the way in which firms are locally concentrated (importance of different industries) and the cohesive nature of IP management in the inventor network, which can be a result of a very incremental and long-lasting clustering process [13]. Despite the diversity of industry-specific conditions, the process clusters must follow to manage their IP may be very similar, and IPR-based educational activities at the cluster level appear to be at stake [15,128]. The case studies of Kranjac et al. deal with two agriculture and agro-food clusters that are geographically close but at the border of different countries [15]. A membership and a management team is set up at the cluster level. Within these clusters, entrepreneurs are non-high-tech. They use traditional knowledge but in a modern way, including tourism activities and implementing ecological principles. Although they have no patents, some have developed forms of IP along with trademark and geographical indication, know-how in unwritten and written forms, business methods and design solutions (protected or not). However, most of them have experienced the violation of IP (50%) and agreed to develop collective IP (transnational level), share IP (2 or 3 SMEs) and individual IP, to better collaborate at the cluster and cross-border level. Corral de Zubielqui, Jones, and Lester also converge to show the difficulties of SMEs to balance IP management and cooperation

for innovativeness [129]. They surveyed 838 networked SMEs of different industries in the Adelaide (Australia) metropolitan area. They mainly found evidence of an interaction pattern where SMEs source innovative knowledge within client–supplier relationships, in particular for commercialization/production-related purposes, as part of a package with the purchase of new equipment or technology. They also find that co-opetition reduces innovativeness, which converges with other findings [130–132]. They put forward the fear of SMEs in this area to give away their technology to a competitor, which could explain their inertia to build new structures of distributed knowledge enabling new paths for innovativeness.

All these studies show two limits to better understanding how SMEs adapt within regional clusters to an open innovation paradigm within an asymmetric IP regime. Firstly, they mainly focus on the MAR (Marshall–Arrow–Romer) model of externalities and on specialized clusters at a particular stage of production within an industry [133–135]. This model might not reflect most regional cluster dynamics [136,137]. Other authors have coined the term 'regional clusters' to focus on the linkages and interdependencies among actors to produce a range of new or re-engineered products and services oriented toward cross-sectoral value creation, while being anchored within the specific socio-institutional characteristics of a locality [129,138,139]. This definition of 'regional clusters', without ignoring the possibility of MAR externalities at a regional level, take into account Jacob's theory about the effect of localized knowledge spillovers on innovation. He assumes that local economic diversity enhances interactions between individuals who have incorporated different knowledge bases [140]. Hence, regional resilience can go beyond path dependency by relying on the ability to innovate while changing the spatial and structural properties of local inter-organizational knowledge networks [136,141–146]. In this perspective, the coexistence of different industrial clusters within a territory could contradict the view according to which SMEs would prefer to scout within their known industrial boundaries [24,102,103]. They can be open to new opportunities for knowledge mobility and new knowledge combinations at a cross-sectoral level [28,29,147–149].

As entrepreneurial experimentation is particularly important for transforming unrelated variety into growth, this could also facilitate the innovation appropriation of SMEs through the structural transformation of the knowledge matrix within cognitive proximities enabled by cooperative relationships at the regional clusters level [150–153].

Secondly, for most of these studies in which firms and their strategies are formally incorporated and tested, the role of firms in the innovation process is still incompletely defined; as it is, the assessment of the underlying factors of collaboration refer to the societal background, i.e., the shared languages, common social norms and cognitive configurations within which such firms are embedded [31,153–160]. Consequently, an SME's ability to balance IP management and cooperation to enhance innovativeness must be understood within a framework in which the types of networks available and the formality of relationships depend on the institutional and cultural context in which SMEs operate [110,136,137,153,161–164]. To overcome these limits, the use of a mixed method with ethnographic observations and focused interviews could offer a richer understanding of the supplier SMEs' learning within regional clusters to both innovate and appropriate innovation.

Our third proposition put forwards that, by trying to create highly appropriable innovation, clustered supplier SMEs can use unrelated regional variety to build cross-business activities and by doing so transform the cluster dynamics from their industrial background.

## 3. Method

The method design is part of the current trend of multi-case qualitative methods where a seminal work suggests that, given the choice and resources, multiple case designs may be preferable to single case designs in a replication logic [165]. In the author's thought this is because each case should be carefully selected either to predict similar outcomes or to predict contrasting outcomes for predictable reasons linked to the theoretical framework,

thereby enhancing the external validity of the study compared to a single case study [163]. However, phenomena such as organizational change or learning extend over time and space, are neither linear nor singular, they involve multiple causal bundles [166]. It is therefore relevant to try to understand them not only at the level of a global case, but also from the detailed study of different units, often intertwined, within this case [165–168]. The longitudinal study of embedded cases thus seems appropriate for a detailed account of complex organizational processes. However, a longitudinal study does not by itself account for the possibility of generalizing these changes in the innovation mode of low- and medium-tech SMES at an international level, hence the interest of a comparison between countries, based on two regional clusters, in France (Provence cluster) and in Spain (Valencian cluster). We chose these two clusters because in both cases the strategy of SME suppliers used to reconcile innovation protection and external cooperation has focused on serving different sectors and markets (within and outside the territory). This coincidence, despite the main customers technological differences (the fresh fruit and vegetable sector versus the aviation sector), had a potential to give some general explanatory significance to the referred strategy of the SME suppliers and this was an important comparison factor. A second motivation for the comparison was to compare the way this strategy was organized in two very different realities. On the one hand, the Valencian reality with a cluster inserted in a region where industrial and agri-food districts with a strong endogenous base are predominant, with a highly developed social capital and a regional government with a broad political and economic autonomy since the mid-1980s. Additionally, on the other hand, a cluster that was built according to a polarized model of state investment in large companies in order to develop declining rural territories at the periphery. In this second cluster, cooperation processes are markedly top-down. They are fostered by a mix of centralized industrial and innovation policy and disparate regional initiatives, all of which resulting in a fragmented relationship between national and internationalizing large companies and the grass root level SMEs with their innovators.

In this international comparative perspective, the industrial dynamic and survival of territories are at the core of the analysis. Indeed, we will consider that the emergence of new firms and networks between enterprises and other innovation centers is essential for competitiveness and, consequently, for the economic survival of the territories insofar as it defines the possibility of generating production and employment opportunities and attracting talent to this territory. Thus, conducting embedded case studies appears to be a necessity to the extent it is not only the individual dynamics of firms that are taken into account but also their collective dynamics and its territorial effects by restructuring industrial relationships between the local and the global. In our methodological framework suitable for comparative regional studies, the company is a first unit of analysis taken in a territory as a second unit of analysis embedding the first one and which has its own history.

Suitable for making comparisons between different units of analysis, this design is consistent with a heterodox grounded theory [169,170]. This one is open to abduction as being a "mean-of-inferencing" to reach to the sphere of deep insight and new knowledge, by identifying and explaining dynamics, patterns from the careful study of a particular contexts [171] (p. 216).

The qualitative study has run over a long period, 6 years for the French regional cluster and 8 years for the Spanish regional cluster. This study over time on the dynamics of knowledge within a population of low-medium tech suppliers SMEs in two regional clusters has enabled us to identify the critical problems leading to revisit entrepreneurs and question them more accurately on the evolution of intellectual property management in their SMEs. We designed the research issue of this latest wave of research in the continuity of the first one. The objective is to understand how the owner managers of these firms through interactions over time have organized their business and cooperation in order to protect their knowledge related to technological innovation within an open innovation paradigm.

Multi-case longitudinal studies tend to be carried out by researchers working in teams, as this requires time and resources that are often beyond the means of an independent researcher [165]. Although we worked in the first wave with a research team, the second wave was carried out solely by ourselves thanks to our easy access to the ground linked to reasons of belonging to the territory (in the Spanish case) or to the governance of the cluster (in the French case).

While in the first waves of research, we used mainly semi-structured interviews to collect the data, we used a mixed methodology, combining semi-structured interviews with a more ethnographical approach over the long term through the immersion of the two authors in professional meetings, speed dating and convivial moments such as business lunches. In the French case, the author has been a member of the cluster governance and more precisely served as Treasurer on the Executive Board between 2009 and 2012. In the Spanish case, the researcher has an easy access because of the length of his research practice in the studied territory.

This repeated immersion on the ground has make it easier to obtain confidential data and to link them to the interviews. It is justified by the fact that the empirical data on organizational practices and strategies in terms of IP protection, notably those of entrepreneurs, remains very tacit. This work of observation encourages entrepreneurs to trust researchers, who must prove that they are not acting opportunistically. This trust is mainly built through long-term investment in the field.

The data collected for this research are therefore of two types: transcriptions of semi-structured interviews and notes taken during formal meetings and less formal talks in convivial locations (restaurant, café, forums, breaks during conferences, general assemblies of associations, etc.).

The semi-structured interviews were led according the method that consists of starting from a main theme displayed in Appendix A.1, we want the person to speak about, and during the interview to rephrase the interviewee's words to obtain more information without giving the researcher's own conceptual categories

They were recorded and transcribed, except for a minority of entrepreneurs who refused to be recorded; in their great majority they were held in both clusters between late 2008 and early 2014. Table A1 in the Appendix B shows the characteristics of enterprises and dates of interview for the Spanish case and Table A2 in Appendix B, shows the characteristics of enterprises and date of the interviews in the French case.

The interviews and the collection of data were conducted along three lines of complementary sub-themes to make the rephrasing, in order to shed light on the phenomenon under study without influencing the interviewee with our categories. The first sub-theme was about their experience of IP protection/theft, striving to identify strong (patents, contracts, trademarks) or weak (lead time, secrecy, design based on clients' specific needs, etc.) mechanisms of protection. We then rephrased sub-themes concerning the strategies and modes of organization implemented to innovate, as well as the modes of external relations (clients, partners), obtaining explanations of the learning developed in reaction to the first experiences, which may have been negative within given national or international legislative frameworks. We also rephrased when they spoke about their networks of solidarity and cooperation in their region as well as their professional or geographic origin. In the Appendix A.2, we present the interview subthemes we used to conduct these in-depth semi-structured interviews with a particular focus on IP protection practices and cooperation between actors in an open innovation context.

At the agro-food cluster in Spain, we interviewed five directors of large firms and eight directors of supplier SMEs (less than 250 employees), as well as seven players from other organizations, including a union delegate for company employees, two union managers, a manager and two inventors of a client (a fruit processing and packaging company) of the supplier's firms, and two legal consultants specializing in intellectual property, i.e., a total of 20 interviews.

In the French aviation cluster, we interviewed three directors of large firms, and 11 directors of supplier SMEs (less than 250 employees) belonging to the aviation cluster, a regional union delegate, an employee in charge of client–supplier relations in the 'competitive cluster' and two lawyers specializing in IP for the Tribunal of Aix-en-Provence, i.e., a total of 21 interviews.

To understand how the managers of SMEs protect their intellectual property over time in an evolving paradoxical regime of strong rights and open innovation, we specify the industrial and societal contexts of the two regional territories.

## 4. Industrial and Societal Contexts

*4.1. The Industrial Sector of the Valencian Community: A Reality Structured by Various Predominantly Endogenous Clusters*

Spain is the world's 5th largest supplier of citrus fruit and the leading exporter of fresh citrus products [172]. This position is due to the specialization of the Valencian Community (VC), which represents 55% of Spanish production.

The VC or Valencian Region is structured in clusters and industrial districts (fruit, ceramics, plastics, footwear, etc.) concentrated in certain territories and which face specific problems [173].

Historically, the Valencian citrus sector has developed technical and commercial expertise that gives it a strong market orientation [174,175]. This sector, and the fruit and vegetable sector in general, stretches along a wide coastal band of the VC. This territory also contains an important part of Valencia's industry, where the suppliers SMEs of the fruit and vegetable sector are located near other clusters within the same territory. This spatial pattern of economic development offers potential interactions between clusters of different sectors both in and between neighboring spaces [146], in the sense suggested by the third proposition.

During the 1980s and 90s, the Valencian regional government (through the IMP-IVA—the Institute of Small and Medium Enterprises of Valencia) developed, in collaboration with professional and union associations, an industrial and innovation policy focused on the creation of technological centers. These technological centers specialized in each of the traditional industrial sectors of Valencian industry (textiles, ceramics, footwear, furniture, toys, etc.), dedicated to research and services for the firms of each of these sectors. Moreover, these technological centers were located in areas where the tissue of SMEs was dense (industrial districts or clusters) due to the concentration of the activity of these sectors [176]. This innovation policy and the SMEs' own strategy have favored the development of a population of supplier SMEs with a greater capacity for innovation and adaptation to customer needs, aided by the social capital of trust that favors cooperation. All of this is in line with what was postulated in our second proposition. During the last decade, there has been a certain tendency towards interaction between the clusters, notably through the scientific and technological influence of the Valencia metropolitan area and the interaction of certain firms with a wider range of technological centers [146].

Since the beginning of the 1990s, the fruit and vegetable sector has had to face new demands from the supermarket chains that now dominate food distribution, as well as the legal limits imposed on the use of chemical products and waste. All this has generated new standards of quality, safety and waste management in fruit processing and packaging companies, which have imposed new requirements on their suppliers of machines and post-harvest chemical products. This process has required and has led to greater scientific and technological development in low- and medium-tech industries such as the machinery and post-harvest chemicals sectors. During the last 10 years, these requirements have led to the introduction of new technologies for the management of production flows [177]. In this sense, according to the interviews, the firms consider the important changes to be the following: concerning post-harvest products, the introduction of low-toxicity products and additives, while new methods for the application of these same products improve safety. As for machines, the systematic introduction of electronics, information technology and robotics has led to the improvement of the fruit calibration system (extension, weight

and detection of the external and internal quality of the product) and the automation of processes such as packing and palletization. All this has led to the implementation of a modern industrial process that meets standards.

This process, combined with the requirements of the internationalization of the sector and its own economic crises, has reinforced the restructuring of the firms in the sector. During the last decade, the two largest firms in the post-harvest machine supplier sector have been acquired by multinationals. The post-harvest chemical supplier sector has also undergone major restructuring, with the merger of national companies and the arrival of certain multinationals. The two principal motors of this process are the investment necessary for innovation and the necessity to open up to international markets.

During this same period, the SMEs of the post-harvest machine suppliers sector developed to meet the same challenge, but in a different way, as they must at the same time meet the challenges set by the strategies of large firms within a regulatory framework for intellectual property that is disadvantageous for them. Such a context, and in particular the increase in size of some large firms and their aggressive IP strategy, may have forced SMEs to resort to more informal means of IP protection, as outlined in our first proposition.

It is important to highlight that the new regional government which emerged from the 2015 elections has promoted (through the IVACE—Valencian Institute for Business Competitiveness—formerly IMPIVA) the implementation of a Strategic Plan for the Valencian Industry 2018–2023 (PEIV). In line with the aforementioned structure of the regional industry—clusters and districts with specific problems—this is a bottom-up industrial policy. The diagnosis of the industry and the actions contemplated in the PEIV are organized around Strategic Challenges and Sector Plans. The former pursue "the improvement of the structural conditions of the competitiveness of the Valencian industry, while the Sector Plans are roadmaps to improve each specific sector/value chain's competitiveness". In the discussion we will compare some of this policy's results with the process studied in this article.

*4.2. The Industrial Sector in the Provence Region: From Development Hubs to Large Company and SME Local Clusters*

Aviation is the leading industrial activity in the Provence Region of France. The sector employs 35,000 persons for EUR 5.5 billion in turnover [178]. According to this source, 1500 SMEs are active in this sector, which, in the region, numbers 10 European-scale clients, four of which are world players. The Provence-Alpes-Côte d'Azur (PACA) region aviation cluster was organized in the 20 years following the Second World War, with the establishment of nationalized companies working for the Ministry of Defense (aviation, nuclear and naval). In the 1980s and 90s, they were restructured several times, with waves of spin-offs [179].

In the 2000s, the new French industrial policy institutionalized the existing regional clusters, generally offering funding to non-profit federating structures which provided them with specific services [180]. The federating structure of the 'Pégase' aviation cluster, created in 2007, is dedicated to services for the local aviation and space industry. 'Competitive clusters' are defined as groups of enterprises, higher learning establishments and public and private research bodies within the same territory that work in synergy to implement economic development projects for innovation. The cluster unit in question is defined around aviation, more specifically helicopters and other flying devices (drones, dirigibles, etc., other than airplanes).

In terms of market position, the cluster includes the world leader in helicopters, world players for the satellite sector and the first French test center. Its main objective is to create a network of large companies, SMEs and research bodies in order to diversify the aviation industry in the PACA region by finding new focuses for development to counter the risk of delocalization of the main clients. At the same time, the institutionalized organization of the cluster as a non-profit association ('competitive cluster') aims to federate innovative enterprises, or those that wish to innovate, around collaborative projects for the aviation sector. In a cluster, the constraint of geographical proximity is justified by the

main clients' specific innovation and development needs for their in-house design and development departments. However, at the same time, the main client focusing on the sale of complete fleets in the area has implemented a low-cost international subcontracting policy, along with the rationalization of their purchasing policy. Given this context, SMEs have experienced recurring challenges negotiating their intellectual property with large companies regarding collaborative research projects [81]. Clustered SMEs have even shown a tendency to collaborate more with public research than with their clients within the clusters as regards innovation [181]. This suggests an asymmetric relationship in the management of intellectual property in line with our first proposition.

Such highly stratified commercial relationships between companies in vertical networks are embedded in social norms where large firm engineers—system architects—and engineers and technicians working in SMEs are not socialized and educated in the same places and professional spaces [182]. However, starting the late 2000s, with the changes in training programs and the new legitimacy of entrepreneurship, low- and medium-technology subcontracting SMEs began to grow increasingly independent vis-à-vis the major principals (aeronautics, as well as naval and nuclear). They used geographical proximity to progress in design activities and aimed at a better valorization of their intellectual assets, whether in different sectors or internationally [81]. Their insertion in the wider regional network of institutionalized clusters led them to new cognitive exchanges with other industries, potentially influencing their choices both in terms of the design and logistics of modular units and of external organization, which ties in with our third proposition.

Technological and organizational modularity is therefore inseparable and is accompanied by a redefinition of expertise—the architect must preserve a 'surplus' of expertise related to their role as integrator [183]. Within the PACA aviation cluster, this modular integration is also based on sharing the standards (aviation certification) required throughout the chain. This evolution, which deeply restructures knowledge, is necessary for 'classic' mechanics and plays a role in de-structuring the frontiers of the industry. It should be added that, unlike 'classic' aircraft, the digital revolution also affects the potential uses of helicopters (or drones). Specialized onboard IT or optical equipment is exploited for new uses both in the aviation sectors and in other sectors: surveillance, the mapping of forest fire risks, diagnostics of engineering works, health diagnosis equipment, etc. [184]. Such a re-composition of knowledge from close relationships with clients' needs from different industries associated to the development of internal multidisciplinary knowledge echoes with our second proposition.

## 5. Results

We first describe the pitfalls and disadvantages that SMEs encounter for the protection of their intellectual property based on strong IP rights, in relation to large firms, even if they approach it differently.

We then study the organizational transformations related to the weak IP protection practices consisting of secrecy, lead time and design in contact with the client within a paradigm of open innovation and technological change.

Finally, we study the evolution of regional clusters based on the strategies of supplier SMEs with high-level technical knowledge creating unique and appropriable solutions, based on the economic and social characteristics of the territory, respectively of the related and non-related variety, and of the nature of the social cohesion in the two territories studied.

### 5.1. Large Firm Offensive Strategies and SME Defensive Strategies

5.1.1. The Strong IP Strategies of Large Firms and the Disadvantage for SMEs

Before analyzing the disadvantages SMEs face when it comes to protect their IP, based on sharing IP rights, in relation to large firms, it may be useful to give a detailed explanation of the content and the general results shown in Tables 1 and 2. These tables show the use made by large firms and SMEs in the Comunitat Valenciana, Spain (Table 1) and

Provence, France (Table 2) of the different mechanisms of strong (patents, contracts and trademarks) and weak IP protection (lead time, secret and innovation based on the client's specific needs).

**Table 1.** Type of IP protection, Spanish agro-fruit cluster.

| IP Legal Content | | Type of Enterprise (1) | |
|---|---|---|---|
| | | Large Firms (5) | SME (8) |
| Strong IP protection | Patents | 1, 2, 8, 9, 11 | 3, 5, 6, 10, 12, 13 |
| | Contracts | 1, 2, 8, 9, 11 | 3, 4, 5, 6, 10, 12, 13 |
| | Trademarks | 1, 2, 8, 9, 11 | 5, 6, 10, 12, 13 |
| Weak IP protection | Lead time, speed | 1, 2, 8,9, 11 | 3, 4, 5, 6, 7, 10, 12, 13 |
| | Secrecy | 1, 8, 9, 11 | 4, 5, 12 |
| | Innovation based on the client's specific needs | | 3, 4, 5, 6, 7 |

**Table 2.** Type of IP protection, French aviation cluster.

| IP Legal Content | | Type of Enterprise (1) | |
|---|---|---|---|
| | | Large Firms (3) | SME (11) |
| Strong IP protection | Patents | 1, 2, 3 | 4, 5, 6, 7, 8, 9, 11, 12, 13 |
| | Contracts | 1, 2, 3 | 4, 5, 6, 7, 8, 9, 10, 12, 13, 14 |
| | Trademarks | 1, 2, 3 | 4, 5, 6, 7, 8, 9, 10, 11 |
| Weak IP protection | Lead time, speed | 3 | 4, 5, 6, 7, 8, 9, 10, 12, 13, 14 |
| | Secrecy | 1, 2, 3 | 4, 5, 6, 7, 8, 9, 11, 12, 13, 14 |
| | Innovation based on the client's specific needs | | 4, 5, 6, 7, 8, 9, 11, 12, |

[1] The number of firms composing the sample of each type of enterprise is shown in brackets (source: the authors, based on the interview guide by Kitching and Blackburn [16]).

In the case of Spain, five large firms (1, 2, 8, 9 and 11) and eight SMEs (3, 4, 5, 6, 7, 10, 12, 13) were interviewed. In the case of France, three large companies (1, 2 and 3) and 11 SMEs (4, 5, 6, 7, 8, 9, 10, 11, 12, 13 and 14) were interviewed. In presenting the results, we will refer to the different Spanish firms with the letter E (for Spain in Spanish) followed by the number identifying each firm. We will refer to the different French companies with the letter F followed by the number identifying each firm. While maintaining their anonymity, Tables 1 and 2 show some salient characteristics of the Spanish and French firms, respectively, which have been repeatedly interviewed.

In both the Spanish and French cases, large firms and SMEs use the same strong protection mechanisms, as a principle. However, as we will see below, they do so for different reasons and to different degrees. Let us say, for example, that large firms make greater use of patents than SMEs and have a much higher number of patents than SMEs, although such a quantitative aspect is outside the scope of this research.

In the case of weak protection mechanisms, in Spain there seems to be an apparent asymmetry between large companies and SMEs regarding the use of lead time and secrecy. However, some firms which produce post-harvest chemical products and machinery for applying the latter products, in particular E10 and E13, do not indicate secrecy as a form of IP protection, due to the difficulties this would entail with regard to the compulsory registration of new products to obtain marketing authorizations. Moreover, some SMEs in the machinery sub-sector, in particular E3, E6 and E7, have so little confidence in the effectiveness of secrecy as a protection mechanism (for reasons that will be discussed later) that they do not even mention it.

The difference between large companies and SMEs is the key role played in the latter by innovation based on the adaptation to specific customer needs, seen as the most effective form of IP and innovation protection. In the Spanish case, the exception, again, is marked by the firms supplying post-harvest chemical products (in particular, E10, E12 and E13). In this specific case, supply is critical in the generation of innovations, for reasons we will point out later, lead time being the key factor of weak IP protection.

In the case of France, innovation based on customer needs also sets SMEs apart from large companies. However, as opposed to the Spanish companies, there is a certain apparent symmetry in the use of secrecy between large firms and SMEs, with SMEs using lead time, while large companies hardly use it at all. These differences between the French and Spanish regions can be explained by a sectorial and a societal effect in the territory. Firstly, in the aeronautical sector, there is a strong tradition of secrecy which is institutionalized through the legislation of the "national defense secret" as defined in the French criminal code and modified by law no 2009-928 of 29 July 2009. The three major companies (F1 for aviation; F2 and F3 for aviation, nuclear and naval) are concerned by this legislation, insofar as they produce processes, objects, documents, information, computer networks and computerized data of interest to national defense which are subjected to classification measures designed to restrict their distribution or access. The French low- and medium-tech SMEs studied here are not concerned by this legislation insofar as they supply elements or sub-systems which are not strategic to national defense. Hence the interest in understanding their use of secrecy by comparing them with Spanish SMEs.

Secondly, the intensive use of lead time in the French LMT SMEs, compared to large French companies, can be understood through the traditional societal relations between principals anchored regionally through a national policy and grassroots subcontracting SMEs. Due to the strong social and professional stratification in French regions, local or regional low- and medium-tech SMEs remained isolated from public research until late 2000 [182]. They supplied the locally established production plants of large firms as atomized industrial sub-contractors. This situation has been referred to as inter-firm Taylorism, where the large firm gives very detailed specifications for the job that needs to be done, and the subcontractor executes the instructions. Nevertheless, we highlighted the ingenuity of low- and medium-tech SMEs, based on their cumulative experiential knowledge, in adapting the design of subsystems with smart discoveries that are integrated to their products without consideration of the purchasing price or of sharing intellectual property [81]. We thus understand that the accessible way for these SMEs to valorize these new inventions, when competing with other subcontractors who may be localized in other countries, is to offer them to other locally established customers as early as possible, in particular in other sectors. In addition, there is a historical difference between the two large companies (F1 and F2) and the company F3 that may help to understand the weak lead time practice regarding the former. While F1 and F2 were in the past nationalized companies with a monopoly in the defense sector, F3 was a low- and medium-tech SME that was acquired by a foreign group in the 2000s. This SME was composed of a valve manufacturing part (which led to a patent registration) and a mechanical machining part.

In Tables 1 and 2, the indicators of use of strong IP protection via patents show that most of SMEs likely to patent do so (6/8 in the Spanish case and 9/11 in the French case), and in terms of contracts, we do not observe a major asymmetry with large firms. The main difference is that large firms are not involved in innovation based on the client's specific needs. These observations are relevant for the two regional territories and need to be analyzed with qualitative data to reveal what processes are at stake behind these numbers, to answer to our three propositions.

In the Spanish, like in the French territories, large companies or at least some of them, are recognized for generally having very offensive strategies for both patents and contracts, which may disadvantage SMEs more than other large companies. The interviews in Spain suggest that SMEs are aware of this asymmetry to benefit from the law framework.

> "If you look at the patents registered with the Patent Office by large companies, 99% of them are worthless. But if you do something, you face 100 patents that you can't defend yourself against even if you are right". (SME E7)

On the French side, we observe, based on jurisprudence that the No. 1 large firm obtained punitive damages against a large Canadian aviation company for a patent copy that was judged to be deliberate, even though the accused company had in fact marketed variants based on this patent. Conversely, in the case of a trial opposing this same company (F1) against another large world aviation company, the court concluded that 15 of the 16 claims of the patent were invalid due to the absence of demonstrated utility and excessive scope.

In our two fields of study, the SMEs criticize the large companies of the sector for using the possibilities offered by the law against the SMEs for their own benefit.

> "If someone wants to see machine X of company Y [SME], they can go to the stand of company Z [large company], which has exactly the same one and yet the first has done nothing against the second". (SME E5)

> "A year later, we came across our exhaust system on a … [foreign country purchasing the technology] military drone. We didn't want to be pushed around. At the request of the police intelligence department, we had a bailiff certify the fact". (SME F12)

In the field of contracts and confidentiality clauses, the problems seem more acute. It is when they are preparing projects that supplier SMEs seem to be the most vulnerable. The large firm tends to appropriate the IP for the design of the original systems proposed by the smaller firms that seek to interest their client:

> "We have recently had two problems with clients. There is very substantial pre-study work. Even though the pre-study is protected, it ends up with the competition … and that's the problem. We are studying methods concerning this problem. The difficulty is to always say enough to gain their trust but not too much, because if we do they use our studies to develop it in-house or it goes to the competition". (SME F7)

We highlight that in Spain, in the case of supplier SMEs for post-harvest chemical products and the equipment to apply chemical products, adaptation to clients' needs is not such an important source of protection for innovations as it is for the SMEs of the machinery sector (cf. Table 1); it is lead time that becomes the key factor. A first explanation is the lack of protection due to the exposure of the codified chemical formula in the patent registration. However, the interviews of the chemical product suppliers give another explanation: the lack of consideration by the clients (packing plants) due to the deficit of scientific and technical knowledge of the latter, who do not appreciate the true value of the proposed innovations and often reveal them to potential competitors (types of wax, etc.). To a lesser extent, this practice also affects the distribution of installation plans designed by the supplier SMEs of the machinery sector.

> "We've had the experience of supplying a client with a sample of a new type of wax and the client passed it on to our competitors. That is why we have to surpass our own innovations. Because in addition … finally, legal protection doesn't protect you that much, you have to give so many clues about the innovation that those who have enough knowledge can do the same with alternative methods". (SME E10)

For all these SMEs in both Spain and France, the verbatim reports highlight the feeling of a lack of legitimacy of their innovation function in the region with respect to more powerful competitors but also with respect to the large client firms that allow themselves to reveal plans, drawings and formulae.

Faced with strong IP standards that are not very egalitarian, SMEs must first develop a usage of the patent that is above all defensive and promotional, as well as other specific

strategies to protect themselves against the risks associated with unofficial asymmetric practices in the contracts.

5.1.2. Constrained Usage of Strong IP in SMEs

An international patent claim and registration represents a cost of around EUR 47,300, not counting annual maintenance costs, which increase with the number of years. In spite of these costs, in France, 8 SMEs out of 13 have a patent. Similarly, in Spain, 6 SMEs out of 8 have registered a patent. The verbatim records did not stress the problem of the cost of registering a patent, the procedure for which has been considerably simplified in Spain with the new 1998 law. Instead, it is the defense procedures that seem to pose a problem. In the case of Spain, the new law appears to generate information and defense costs that are more difficult to support for an SME than for a large firm.

"And if you consider that you have been prejudiced, it is not enough to report the situation to the Patent and Trademark Office for them to check the originality of the innovation and take sanctions. You have to take the matter to court". (SME E3)

In France, defending a patent is also more expensive, dissuasive even, for an SME.

"Once it [patent infringement] was confirmed, I went to a bureau who told me to drop it because it was T . . . [Name of the French multinational that had copied it] and I . . . . [The nation state that purchased the technology], and it would cost us 5000 euros in fees to sue them". (SME F12)

In fact, in both countries, SMEs mainly register patents out of fear of not being able to exploit their own invention because it may be patented by a competitor.

Protection is above all to prevent the large company from swallowing the SME (Employee of a Spanish IP agent's bureau, 2014).

In France, those SMEs that develop sub-systems enabling them to become essential in their field of expertise also have a strategy of selective patent registration that aims to ensure they grow autonomously at the national and international levels with respect to their main clients.

"We produce our products, we have resources and we block our strategy very well by patenting prototypes to protect ourselves ( . . . ). A patent research means the protection of know-how, yet we are not protected for 70% of our products". (SME F6)

In the case of Spain, with a law that authorizes registration without verification of anteriority, patents are seen as a means of protecting the client to which the innovation has been sold. In Spain, through the 1986 Patent Law the grant of a patent granting became conditional on the demonstration of the novelty of the innovation. However, the Royal Decree-Law 8/1998 on emergency industrial property measures introduced a second system for granting patents that did not require prior examination of novelty and invention by the Spanish Patent and Trademark Office. This procedure has favored the blocking of SME innovation by large companies. The situation only changed very recently with the enforcement on 1 April 2017 of the Patent Law 24/2015, which reestablished the prior novelty and invention examination step as the only system for granting patents.

"Protection also offers guarantees to the client of the SME, to avoid them having to stop using the machine two months later due to a patent infringement registered by a larger holder". (SME E4)

The SMEs of both countries resort to contracts, offensively or defensively depending on the nature of the relations between clients, suppliers or partners. In Spain, we observe an offensive use of contracts by purchasers in the machinery and post-harvest chemical products sectors. These impose exclusivity and confidentiality clauses on their suppliers. SMEs that resort to sub-contractors for special parts reproduce this behavior by appropriating the supplier's creativity.

"If we order a specific (made to measure) part for ourselves, it is obviously a product that we would want to protect". (SME E4)

In the case of France, the offensive use of contracts by SME equipment manufacturers is also developing with respect to their clients.

"We developed products . . . With . . . [purchaser in the aviation sector], we signed the contract, development was exclusively carried out in-house according to this contract . . . we maintained our independence: they gave us their specifications—a sort of sketch! There was nothing legible on it . . . it was up to us to find the simplest thing possible; we went the INPI and registered a patent—we were protected to transfer the innovation by maintaining the patents". (SME F6)

*5.2. Entrepreneurial Creativity and Cross-Business Activities in the Clusters*

In both clusters, the offensive use of patents and contracts by large companies leads SMEs to elaborate combined strategies of secrecy, lead time and complex design that transform their relations within the cluster.

5.2.1. An Integration of Multidisciplinary Know-How for Generic Competence That Is Difficult to Imitate

Supplier SMEs have learned to defend their IP through secrecy with the integration of new knowledge (automation, IT, sensor technology, simulation) in the activities of design, prototyping and production leading to the encapsulation (often subtle and without concealing it) of the innovation in the product or service. In the case of Valencia, the integrated development of palletization, automation and robot technologies generates a series of generic skills.

"We are a company that has emerged in an engineering perspective that is more technical than commercial. The company began to become known for the production of palletizers for ceramics [for which they registered several patents], but later there was a diversification and complexification in the fruit and vegetable, agro-food and chemical industry sectors . . . we then developed other types of machines and made progress on the integration of all the automation essentially associated with palletizing". (SME E5)

"Faced with imitation, our answer has been: instead of making a palletizer, we make palletizing robots. Thus we develop a whole series of skills for the integration of machinery and automation". (SME E3)

In Provence, it is the integration of the knowledge of sub-systems or complete technological systems (cockpit dashboards, high-performance hose systems, soundproof partitions using rubber that associates mechanical and chemical expertise, small aircraft dedicated to various sectors) that generates solutions that are difficult to copy.

"There is an interesting thing in our profession: at the beginning there were a few people with some chemical knowledge and with that, around 10 years ago, we ramped up to become a design office, we hired people with materials knowledge and who understand mechanics—to complete a project we needed to conjugate these two aspects, I mean we try to marry chemistry and mechanics—and it's not an easy marriage—because they are people who speak two different languages". (SME F5)

The implementation of secrecy is intimately related to the transmission of knowledge and the skill of integration, which implies a control of the value chain in the niche; the creator manager of the SME is often alone to hold the integrating know-how at the origin of the product/service and the core of the technical knowledge they transfer.

*"The boss was a technician and not a sales person and he liked to meet the challenge of developing things. We developed a very wide range of products and innovations and that enabled us to learn an enormous amount of things.* (SME E5)

*I train the interns as well as new recruits. I am the one who holds the skills and knowledge and who ensures the continuity between design, prototyping and production.* (SME F13)

*Even if we work with sub-contractors, we mustn't give them the total production of a product ( . . . ). We produce our product from A to Z: the engine manufacturer delivers the engines in parts and we assemble the entire engine and machine.* (SME F14)

We note that SME F13 is a spin-off of large company F1, a sales agreement for the light aircraft that F13 designs, develops and has produced was made with F1, which possesses the sales networks and the initial patents for this inventor, which now protects itself through secrecy and their integrating know-how.

In the case of the SMEs of Valencia that do not register patents and use open source, innovation is not really protected insofar as the software programs remain open. However, copying remains difficult for a person who does not have highly developed skills, even more so when the solutions are specifically adapted and proportional to the client's requirements.

"Our innovations are free, they are already in the machine, because I think that the people who buy your machine buys all its devices, and I know that the person who is capable of copying your machine is just as capable of producing it themselves, without seeing your model. That is why, in principle, the programs are in all the machines. You don't make them freely available on the internet, but you don't try to hide them so that no one sees them either". (SME E7)

In spite of these organizational choices, secrecy is not guaranteed for the SMEs, which incites them to a type of organization that supports lead time. The common trait among the SMEs of the two clusters for the organization of lead time is the combination of up-to-date technological know-how with commercial know-how. This combination enables them to convince the client of the value of the technological innovation while at the same time protecting their knowledge.

"All these firms share the common denominator of an innovative and creative segment or niche, which often combines with the commercial dimension and the search for profit". (SME E6)

"We work quite a bit in partnership and collaboration, so the people spend a lot of time on those subjects and in the end we came against a problem of profitability— we have now rationalized things—I take care of the commercial monitoring of the subject, so I intervene at the same time as the engineers—And every time there is a policy or commercial decision to be made, I am there . . . there is a double vision". (SME F5)

The strength of SMEs is to have established their commercial reputation based on a bespoke service that integrates innovation and thus creates technological niches.

"The philosophy of our company is that we must not facilitate the possibility to copy us, but neither must worry about knowing whether we have been copied, because the next machine we make must be more efficient than the previous one". (SME E3)

"In the beginning, we only manufactured the backlit face of the equipment, then we moved on to small cockpit equipment until we offered a complete system. We do not register patents: in our field it is safer to keep the lead . . . we take markets for which they don't know how to make the parts elsewhere. When we propose a system or service, it's much more complex to sub-contract elsewhere". (SME F9)

However, under the pressure of an INPI adviser who acclaimed the merits of patents for marketing purposes, this SME finally registered one in 2015.

### 5.2.2. Appearance of Formal and Informal Cooperative Relationships for Innovation

In the SMEs of both clusters, we see the appearance of a strengthening process of cooperation based on the interactions between formal (contracts) and informal exchanges linked to growing trust between partners.

> "Relations with these companies were formalized in a contract when we began to work with them to commercialize their boxing machines in 2003–2004. But because we had a good relationship with this company, including for our respective property, it meant that our relations were more informal and things were clear: areas and sales prices". (SME E5)

This development of informal relations in the two clusters, related to the difficulty of strong IP protection, can in particular be observed in those SMEs that develop software. We observe that the granting of licenses only exists when trusting relations are established upstream. The case of SME F7 illustrates this:

> "Software protection is very complicated to set up. Today, we don't sell it alone. We sold a few CNS (naval propulsion) licenses to the Nantes site, we which have known for a very long time (purchase of 7 licenses)". (SME F7)

Note that the founders of this SME are IT specialists of the nuclear sector, who performed mechanical tests for clients of this sector and who had the idea to create generic software for test benches for engines (acyclic analysis) designed for different sectors; an idea which did not interest their original company.

In the Valencian Community, thanks to informal cooperation, licenses were purchased by SME E5, established in the ceramics sector, from SMEs who are experts in new technologies, enabling them to diversify their activity into the fruit and vegetable sector.

> "The time came when we made an agreement with a local company that had developed an automatic citrus-bagging machine, to be able to sell this machine throughout almost all of Spain". (SME E5)

These SMEs, which cannot protect their innovation through patents or secrecy, are led to create a cross-business group that meets the specific requirements of clients in the machinery sector. For example, four SMEs from Valencia (including E3, E4, E5 created in the 90s) and two large multinationals from the Netherlands (E8 and E9) collaborate on the construction of a modern fruit and vegetable packing facility. These SMEs have developed a set of complementary new skills, which considerably reduces the possibilities of imitation and demonstrates the appearance of a vital skill which consists in knowing how to cooperate without transferring the company's matrix of knowledge.

> "We are not going to copy general programs that enable the control of the machine (programmable logic controller) that the other company can supply any more than we supply our calibration software, because these programs are confidential to each company". (SME E4)

These observations suggest that the most recent SMEs, on the basis of their multidisciplinary knowledge, create new opportunities for collaboration and learning for those SMEs that have long been rooted with their workshop, within the territory, in a perspective of open innovation. At the same time, these more recent SMEs create the conditions to weave new forms of relations with large multinational companies that market packing plants, and which see in this new type of group opportunities to innovate in one part of the technical facility at the heart of the profession (fruit packing), thus distinguishing themselves from the competition.

In France, this culture of horizontal partnership between SMEs, and moreover with larger competitor companies, is less spontaneous in a territory that is structured by relations of vertical quasi-integration. In all, three SMEs cooperate with other SMEs with complementary skills but in a way that is more formal (SME F5, SME F9) than informal

to innovate (SME F7, F12, F6). The in-house development of generic knowledge for specific equipment at the interface of different sectors is seen as the main solution for the valorization of creativity to confront a lack of trust towards large companies or other SMEs.

> "We try to work on the interface between the user and the product—we have created our own product with a touchscreen interface and a little more electronics (with a PDA [Personal Digital Assistant], plus audio light menus—always for aviation; with this product we hope to be able to move away from aviation!" (SME F9)

At the end of the period, this SME valorized its specific competencies in the health sector by acquiring a license issued from public research for the diagnosis of breast cancer and collaborates with the spin-off created to develop and market this technology, all while acquiring shares in the company. In this territory, this strategy matches the strategy of older SMEs (F5) that have adopted, in order to protect their IP and knowledge, strategies of external growth through the purchase of shares in partner companies while rationalizing their collaboration with other partners.

> "In 2005, we set up a holding with different companies in a sector close to ours. We have a subsidiary that works in the field of composites—we have joint projects with our client [aviation company, defense sector in the region] . . . " (SME F5)

In France, the capacities of these SMEs to create more horizontal networks of cooperation with other companies by limiting themselves to informal strategies or contracts occurs much less spontaneously than in Spain. The strategy of acquiring shares seems to be in correlation with the relations of cooperation on innovative projects. Nevertheless, these new strategies of open innovation on the basis of highly specific in-house knowledge inspire the confidence of the large client companies of the region, which, as in the Valencia region, are beginning to see a reciprocal interest in cooperating with these relatively recent SMEs whose inventiveness they tended to devalue.

The regional aviation employers' federation only began working on the relationships with small turnkey contractors once their clients said that it was important, while [the large aviation company F1] or [the large aerospace company] did not say.

> "these suppliers have the codes of the value chain, they have the technology that can interest the groups, now we can imagine prototyping an idea with F9, for example . . . " (ex-director of R&D with F1, then director of the aviation and space competitiveness cluster in Provence)

This verbatim report suggests that the engineers from major schools (aviation) that concentrate in these large client companies have modified their view of the capacities for innovation of supplier SMEs. Moreover, F9 left F1 to create his own company in association with a local SME workshop in the printing sector because he was refused a promotion to a position that is usually held by an aviation engineer, but which he considered he merited following a well-conducted project; he was a technician and attended a school of engineering through a continuous training program.

## 6. Discussion

Our qualitative results contribute to establish our first proposition, according to which, even in an open innovation paradigm, SMEs show more weaknesses than large firms to protect their knowledge and innovations through strong and formal IP rights. They also lead us to refine this proposition.

As shown in Tables 1 and 2, the majority of supplier SMEs in France and Spain use strong IP rights (patents, confidentiality clauses in contracts and trademarks), but the verbatim reports also reveal a reality that is different from that of large firms. These observations lead us to specify that, in the continuity of the literature, LMT supplier SMEs create patents defensively especially in order not to have to defend themselves or to be prevented from the appropriation of their own innovation by competitors who register the

same patent [19,66,70]. Moreover, the supplier SMEs studied do not register patents mainly with the objective of financing their R&D efforts via income generated by licenses [67]. Supplier SMEs holding patents realize their difficulty to exploit these patents alone insofar as, in contradiction to what is suggested in the literature, patents do not represent a sufficient means to ensure their reputation and marketing to make them profitable [66,70]. It is by cooperating with these preferred SME partners that they envisage the possibility of developing licensing activities, even if, at the origin, they did not have this ambition. With regard to contracts, in the clusters studied, the large companies that use clauses of confidentiality and exclusivity asymmetrically, and which do not recognize the creativity of a supplier SME, will influence the formal IP protection behavior of these SMEs. The SME may rebel by demanding compensation for its knowledge (Spain) or by registering a patent without the main contractor (France) knowing of it. One can suppose that the power balance will critically depend on the more or less strategic character for the main contractors of the knowledge transmitted by the SME. In conclusion, for the supplier SMEs studied, achieving open innovation while protecting a part of their IP with the mechanism of strong IP is very difficult. The same applies with contracts that contain unilateral IP clauses, which lead them to only use strong IP rights in a defensive way. These differences and similarities at the level of the first proposition are synthetized in Table 3 below.

**Table 3.** Comparison between the SMES strategies in the Comunitat Valenciana and in Provence (micro-level).

| Propositions | Comparison between the Comunitat Valenciana (Spain) and Provence (France) | |
| --- | --- | --- |
| | **Similarities** | **Differences** |
| First proposition: SMEs show more weaknesses than large firms in terms of knowledge and innovation protection through strong and formal IP rights (micro-level) | SMEs use patents and contracts defensively and not as a means to finance and protect their innovations. SMEs protect their innovations essentially by offering solutions tailored to their customers' needs. | In Provence SMEs patent when they design new elements of technological sub-systems, not necessarily asked by clients, in subcontracting relation with larger companies. Valencian SMEs patent when they have a "standard" product that they can market to several clients, but they prefer to protect specific solutions via lead time. |

In line with our second proposition, we show also how LMT supplier SMEs can learn to manage IP through their initial competencies while forming innovation cooperation. First of all, almost all the SMEs studied simultaneously use secrecy, lead time and complex design, which in a certain way reflects the literature. By basing ourselves on an approach in terms of learning, we are able to specify how these supplier SMEs articulate these different informal IP strategies, in contrast to part of the literature [20,21,100]. For these SMEs the implementation of secrecy and lead time are facilitated by a dynamic relationship between the in-house construction of a multidisciplinary technical skill for the integration of systems and the capacity to adapt to the needs of the client. This learning process enables the integration of different technical sub-systems into an original complex design that is difficult to imitate. Above all, it seems that it is via this means and not principally via secrecy and lead time that they protect their innovation. Our interviews and field observation also suggest that this organizational learning and the associated mental model are constructed on a daily basis from the initial failures and deceptions related to low recognition and lack of respect for their inventiveness, especially by the large client firms. Finally, we highlight, within the population studied, a pugnacity to cooperate around innovation, channeled by the capacity to develop commercial, legal and technical expertise, in contrast to those authors who say that they are afraid to cooperate [129]. However, our results above all highlight that the complex design that enables these supplier SMEs to better protect their intellectual property and to cooperate with other partners is founded on in-house technological multidisciplinary knowledge (IT, electro-mechanics, optics, robotics, aviation, etc.) as opposed to functional multidisciplinary knowledge (marketing, management, technique) [100]. These differences and similarities at the level of the second proposition are synthetized in Table 4 below.

**Table 4.** Comparison between the SMES strategies in the Comunitat Valenciana and in Provence (meso-level).

| Propositions | Comparison between the Comunitat Valenciana (Spain) and Provence (France) | |
| --- | --- | --- |
| | Similarities | Differences |
| Second proposition: LMT supplier SMEs grasp ideas mainly from close relationships with clients and their needs in different industries to develop internal multidisciplinary knowledge which they can use as a basis to balance innovation and IP management cooperation, involving formal and informal means of protection. (meso-level) | SMEs are able to develop complex competencies and to cooperate with other SMEs and large companies while protecting their IP. | In the Valencian Region, the intra and inter-sectoral networks of SMEs are part of an historical social capital of sectoral and territorial trust. Large companies (including multinationals) either local and/or rely on local SME delegations in the machinery sector and/or operate with local technical staff in the post-harvest firms. This favors horizontal relations even between large companies and SMEs. In Provence, there is a traditional split between the engineers community of large companies (linked to the state-run "grandes écoles") on the one hand, and technicians and engineers of endogenous SMEs, on the other. Recently, however, some communication spaces and mutual understanding have recently evolved. The creation of institutionalized clusters on the bases of first industrial networks has enhanced these new mediations. |

Our third proposition suggests that, by trying to create highly appropriable innovation, clustered SMEs can use unrelated regional variety to build cross-business activities and thus transform the cluster dynamics from their industrial background. Based on our results, the arguments to advance along the lines of this proposal are structured around three points.

Firstly, the supplier SMEs we studied were created through the mobility of a new and more qualified generation of entrepreneurs who have acquired a knowledge base within clustered industrial firms through a cognitive proximity with their clients. It is mainly by initially performing complex design on their own that they become less reluctant to cooperate in a perspective of open innovation. This observation is close to that of Chesbrough, who discussed the case of a Xerox spin-off that went out on its own to start up a new company through its capacity to combine diverse disciplinary knowledge at a cross-industry level (personal computer and computer networking industry) [76] (p. 9). Our survey shows that this can also happen for LMT supplier SMEs. Thanks to their knowledge base enabling complex design, these SMEs are capable of designing product solutions that are sophisticated technical systems, either on their own or through the integration of the complementary solutions of companies within the tiny networks they have built [185]. By trying to create original and highly appropriable innovation, the CEOs of clustered supplier SMEs use the unrelated variety they can reach in the regional territory. However, it must also be noted that because of public–private initiatives to reinforce the links between research and industry and between industries, innovating at a cross-business level in the territory has become less time wasting. In this respect, we converge with the recent work of Fritsch and Kublina [152] highlighting the positive effect of unrelated variety on growth within regions showing higher levels of new business formation associated with higher levels of absorptive capacities in terms of research and development activities.

Secondly, among the equipment and systems supplier SMEs studied, we showed that strong IP rights are essentially used defensively, while secrecy may be maintained thanks to the expertise of a firm for endogenous complex design. In Spain, this competency nourishes a capacity for cooperation for inter-organizational complex design with international firms that are leaders in the same commercial segment that seek to make their technical systems more powerful and better adapted. These firms are interested in the search for differentiation with regard to those competitors who satisfy themselves with standardized offers of equipment at the international level. These latter remain relatively costly while requiring the local client to adapt. In France, this competency of the supplier SMEs studied extends to the capacity of production of the final product able to compete with the products of international leaders (e.g., drones, including ultralight helicopters, to carry out missions that are difficult for a traditional helicopter and less expensive). One may have expected

very different relations between clients and suppliers according to the specialization of the two regional industrial clusters studied. However, when we look at the creative perspective of equipment supplier SMEs in both regional territories, through their capacity to work for several sectors, our results show that they define, in convergence in the two regional clusters studied, their own conditions for the protection of IP and innovation.

Thirdly, the method of building this relationship of trust is different from one region to the other. Whilst in the Valencia regional territory trust seems more spontaneous, this is not the case in the French territory. To understand the differences in the strategies of cooperation for open innovation, it is not only in related and unrelated variety that we need to seek an answer, but also in the more or less cohesive nature of the social links in these territories with respect to the evolution of qualifications and public policies [31,137,153]. These differences and similarities at the level of the third proposition are synthetized in Table 5 below.

**Table 5.** Comparison between the SMES strategies in the Comunitat Valenciana and in Provence (macro-level).

| Propositions | Comparison between the Comunitat Valenciana (Spain) and Provence (France) | |
| --- | --- | --- |
| | **Similarities** | **Differences** |
| Third proposition: By trying to create highly appropriable innovation, clustered supplier SMEs can use unrelated regional variety to build cross-business activities and by doing so they transform cluster dynamics from their industrial background. (macro-level). | Supplier SMEs have been instrumental in opening new evolution avenues based on a sectorally and spatially integrated regional industry. | In the VC, this process involves the complexification of endogenous-industrial districts, also supported by the main metropolitan areas and an industrial and innovation policy decided by the regional government. In France, a dual structure is being consolidated in the territory, with sectors that are impulsed by state initiative and supported by large companies (nuclear, aviation, etc.) together with an endogenous based industrial and tertiary structure. Industrial and innovation policy is still state-led. Nevertheless, the institutionalization of regional clusters has created spaces for negotiation and mediation for SMEs belonging to one or several different clusters. |

The supplier SMEs studied in the Valencia region demonstrate a greater need to co-operate due to Spain's recent 1998 IP law, which is particularly unfavorable. However, these enterprises benefit from more opportunities for open innovation in a territory where speaking the same local language is an asset for more spontaneous cooperation, considering that this language and the local culture it nourishes fully benefit from the 1982 law that conferred the status of regional autonomy to the Valencian Community. This same law also enables a participative policy of industrial innovation with employers' federations and employees. The SMEs studied in the Valencian Community activate standards of geographical and social proximity existing in the territory in favor of organizational proximity [116,153]. This results in the structuring of relations and horizontal industrial dynamics where an effect of reputation and trusting relations under social control come into play. This confidence is supported by the endogenous character of SMEs and the strong presence on the territory of large companies (see Table 4), which allows for the formation of a dense network of SMEs and SMEs with large companies, which in turn rely on other actors of the local and regional innovation systems.

In Provence, the social tissue is less cohesive for historical reasons that are both regional and national [186,187]. The SMEs are facing with not very dense social relations leading to a type of anonymity due to the fragmented history of the Provence Region, which demonstrates significant stratification and social and geographical discontinuities. These result in an irreversible rural exodus towards the Mediterranean coastal area associated with the initial establishment of clusters inspired by the Colbertist doctrine of Perroux (nuclear, naval, aviation, petrochemical industry). From the 2000s, the development of a regional governance of innovation, with the creation of competitiveness clusters, came to support the territories by opening spaces of sometimes fierce negotiation among SMEs, large companies, scientific and technological research bodies and training institutions.

However, the recent state policy favoring competitiveness clusters has contributed to the generation of new interaction spaces between the two communities of practice and two previously distinct logics (exogenous versus endogenous), which could lead to a process of greater endogenized or territorialized French industry.

In this context, the more protective IP law towards SMEs is that of Spain, that has since evolved in the same direction as the French law. This institutional change was intended to strengthen the growth of the legitimacy of small structures. However, the transformation of the balance of power within the territory leading to a growing dependency of main contractors with respect to complex technical system supplier SMEs is above all linked to their capacity to create based on their endogenous multidisciplinary capacities of innovation at the interface of sectors [87]. Even if these firms begin to develop micro-networks of cooperation with other SMEs or with public research by purchasing patents to develop, the relations of cooperation are less spontaneous than in Valencia. The frequent practice of purchasing firms observed among supplier SMEs, equipment suppliers in the case of Provence, comforts our observations on this difficulty to cooperate.

In summary, we show that the driving role of creative entrepreneurs in the transformation of cluster dynamics may find its source elsewhere than in the rooting of a firm that is exogenous to the territory, as has been demonstrated elsewhere [11]. Supplier SMEs play a driving role in the dynamics of territorial innovation and the differentiation of large companies at the international level, mainly via an entrepreneurship that is rooted in a transformation that is endogenous to the territory. The soil for this evolution jointly consists of the diversity of industrial activities, an evolution of qualifications, public policies promoting both the development of technical and scientific knowledge in these different activities, and localized social negotiation. Part of the literature has insisted on the difficulties for SMEs to protect their intellectual property within a strong regime and another part of the literature has insisted on the difficulty of SMEs to protect their intellectual property in open innovation.

The article establishes a productive dialogue with some of the latest literature on clusters and on innovation drivers in SMEs.

Recent work on various clusters and industrial districts in the Valencian Community has envisaged the possibility that these clusters and/or specific districts might adopt radical innovations (digitalization) supported by the aforementioned bottom-up industrial policy. Such radical innovations would allow them to circumvent lock-in problems through the action of collective actors (technological institutes, etc.) [153,188]. Our article offers a more spontaneous way of achieving modernization and fighting against the lock-in of clusters based on the exploitation of intersectoral interrelationships by SMEs, although surely taking advantage of the opportunities of the bottom-up regional policy.

Our article consolidates and clarifies the literature on innovation drivers in SMEs [189]. Indeed, this is an area where SMEs make certain efforts in terms of R&D and to develop internal skills according to the science and technology-based innovation (STI) model. However, this type of R&D is closely linked to activities other than typical R&D (design) and it hinges on the direct and seamless interactions of R&D department technicians with the company's own customer management. The latter is further away from the STS model and closer to the DUI innovation model (learning-by-doing, by-using and by-interacting) [189]. This opens a research field for a potential improvement pathway for clusters and regions with a predominance of low- and medium-technology sectors. By following such a pathway, the exploitation of intersectoral relationships in the context of regions and territories hosting different types of sectoral clusters would allow SMEs to combine both innovation models, thus contributing to the development of clusters and of the regional economy as a whole. We agree with Hervás-Oliver, that in SMEs collaboration with other actors in the value chain may be more important than R&D. However, in our case, what makes such a collaboration fruitful is not the internal weaknesses of SMEs, but rather the development of their in-house technological integration capabilities.

## 7. Conclusions

We sought to understand how a particular category of SME, LMT supplier SMEs, in a context of evolving clustering processes within multi-specialized regions, learn and organize themselves in a new way to address the challenge of IP protection, within an open innovation paradigm.

The originality of our work is to show that LMT supplier SMEs are capable of transforming the innovation dynamics of the cluster from which they appear while protecting their intellectual property via a cross-sectoral offer of complex innovative technical systems. These, seeking to systematize this type of innovation, manage to cooperate with other SMEs and with large companies interested in strategies of strong differentiation; on the way, they transform the balance of power within their industrial cluster. However, this capacity cannot thrive without fertile soil; it is rooted in territories with the specific particularities of multi-polarity, a high level of education, and territorialized policies of industry and innovation. Finally, the behavior of these firms, which play a fundamental role in innovation via the territorial rooting of value chains, depends more on their relationship with knowledge within the territory that covers several sectors than on a sectoral logic.

Our results shed new light on the regional dynamics of innovation with respect to most of the work on the relationship between IP and open innovation. These works mainly treat high-tech clusters focusing on one or several phases of design, within a given sector, whether it be biotechnology (especially agro-food) or TIC (software and video games). These sectors are based on R&D, for which the legal framework has been adapted to enable profitability based on the appropriation of the living (patent), computer software or video creation (copyright). In contrast, the SMEs studied have no other solution than to develop learning to protect their IP within a regime of strong protection which is unfavorable to them. This situation appears to stimulate open innovation based on inter-sectoral relations within territories consisting in different industrial clusters.

Our work also highlights that in studies focusing on the dynamics of a cluster, the relations with other sectors in the territory may be neglected. Yet, it may be in these cross effects that the innovative dynamics of a territory are to be found, with technical systems and associated services supplier SMEs as the cornerstone.

From a methodological point of view, we do not compare industrial clusters that are a priori not comparable if we suppose a specificity of the coordination of productive activities among enterprises as a function of the market and technological characteristics of this sector. What we do compare are the behaviors of innovation and intellectual property protection of supplier SMEs, originally belonging to the industrial clusters studied but becoming cross-sectoral in their offer of complex technical systems. It is in this latter characteristic that lies the pertinence of our comparison.

Indeed, the most important difference in the behaviors of this type of firm between the two regions, based on the comparison we carried out, is a difference that is above all societal rather than sectoral. What we mean by this statement is that the institutional framework regulates social relations, the most relevant actors play and the specificity of the innovation systems in both countries, and this marks the main differences between them. Specifically: an endogenous industrialization process and specialization in the consumer goods sectors close to final demand, such as in the Valencian Region (Spain), as opposed to the model of state-driven development poles in the Provence Region (France). What is more, the industrial and innovation support policy for local actors in industrial and agri-food districts as defined by the Valencian regional government in a highly decentralized country contrasts with the overall state policy of stimulating competitiveness clusters nationwide. Both societal differences define the basic explanatory matrix of the differences at the micro, meso and macro levels (see Tables 3–5). Interestingly, these major differences have not prevented the strategies of SMEs in both regions and countries from converging towards the formation of a web of cross-sectoral relations, this being a key element for generating innovations and at the same time protecting SMEs in an open innovation context.

Finally, we will present three limits to our work, opening as many paths of research. We did not carry out an in-depth study of the human resources strategy of these firms to more finely understand how they internally construct their capacity to balance open innovation and the protection of intellectual property. A first path of research would thus be to carry out deeper observations of the management of human resources and the management of knowledge in the SMEs of these two regions. Secondly, to consolidate these results, we could study a larger sample of firms in diverse industrial clusters to test the robustness of the results at the level of different regions. Thirdly, one of the limitations of this work is that the empirical part of the paper focuses on the study of the strategies of supplier SMEs in only two European countries and clusters. However, we believe that the results of this research allow us to put forth some wider conclusions for SMEs in the European Union or SMEs globally. Finally, such broader conclusions could be hypotheses that could be tested in future research covering new countries for a broader scope of comparison.

The article provides evidence that supplier SMEs, even in the context of different institutional frameworks and different sectors, are able to define a specific strategy to meet the challenges of IP protection in an open innovation context. As a first general conclusion, this suggests that SME networks in low- and medium-tech sectors are able to find their own way of innovating and protecting their innovations. A second conclusion is that this strategy mobilizes the cross-sectoral integration potential of new technologies in the hands of people with a technical profile who are capable of imagining new solutions and activity niches. This builds on (and leads to) the development of generic competencies in SMEs which allow them to respond creatively to the specific, even unique, requirements of their customers in different sectors. The transformative capacity of the sectors, clusters and territories that drive this process also stands out. In addition to the renewed dynamism of a cluster specializing in one sector, and the possibility of interacting with other sectors through SME suppliers, we are witnessing the progressive integration of value chains. In this respect, the article provides a genuine avenue for developing a smart growth support policy, as regards SMEs operating in low- and medium-tech sectors that have upgraded their technological level both on the supply side (new technologies) and on the demand side (new customer requirements).

**Author Contributions:** Conceptualization, J.R.G.-B. and M.G.; methodology, M.G. and J.R.G.-B.; validation, M.G. and J.R.G.-B.; formal analysis, J.R.G.-B. and M.G.; investigation, M.G. and J.R.G.-B.; resources, J.R.G.-B. and M.G.; data curation, J.R.G.-B. and M.G.; writing—original draft preparation, M.G.; J.R.G.-B. and funding acquisition, J.R.G.-B. and M.G. All authors have read and agreed to the published version of the manuscript.

**Funding:** This research was funded by the Secretary of State for Research (Spain), first grant number DER2011-23528 and second grant number CSO2016-78169-R. The APC was funded by University of Valence and the National Center for Scientific Research.

**Institutional Review Board Statement:** Not applicable.

**Informed Consent Statement:** Informed consent was obtained from all subjects involved in the study.

**Data Availability Statement:** To access the verbatim data, contact the authors by e-mail for an appointment to make the data available.

**Acknowledgments:** We thank André Torre and the attendants to our session at the ASRDLF conference (Association de Sciences Régionales de Langue Française) in July 2018, for their oral comments during the presentation of a preliminary version of this paper. The authors are also grateful for the interesting suggestions of two anonymous reviewers and of the editor of this special issue of Sustainability, who helped us to improve the final result of the article.

**Conflicts of Interest:** The authors declare no conflict of interest. They also declare that they used the alphabetical order of authors' names to sign this paper. The funders had no role in the design of the study; in the collection, analyses, or interpretation of data; in the writing of the manuscript, or in the decision to publish the results.

## Appendix A. Main Theme and Subthemes Addressed in the Interviews Led on the Two Regional Territories

*Appendix A.1. Main Theme Explication with the Interviewees*

"We thank you for hosting us and would like to talk with you about your innovation strategy and practices and how it works with partners, customers or suppliers for intellectual property."

*Appendix A.2. Subthemes to Rephrase the Interviewees Words*

1. General context of companies and innovative dynamics.
2. Motivations to protect innovations and intellectual property (IP).
3. Mechanisms used to protect IP and motivations of each.
4. Tensions and cooperation between large companies and SMEs in the protection of IP.
5. Organization (internal and external to the company) of IP protection.
6. Management and regulation of the IP of SMEs and their employees in the territory.
7. Influence of the protection of intellectual property in the process of business internationalization.
8. Verify if there is a growing technological complexity of the innovations and of the communities of practice and of the epistemic communities involved.
9. Business, sectoral and territorial effects of said complexity.
10. Management and regulation of IP with suppliers, subcontractors, clients and universities and research centers.

## Appendix B. Characteristics of Enterprises and Dates of Interview

**Table A1.** Characteristics of enterprises and dates of interview—Spain.

| Enterprise | Activity | Size | DOMC/SME/SU | Date of Creation | Date of the Interview |
|---|---|---|---|---|---|
| E1 | Manufacture of production line machinery for the fruit and vegetable sector | 1000< | Main contractor and supplier Large multinational firm | 1905 | 18 April 2006 23 April 2014 |
| E2 | Manufacture of production line machinery for the fruit and vegetable sector | 500< | Main contractor and supplier Large multinational firm | 1956 | 9 April 2006 20 May 2013 1 April 2014 |
| E3 | Automation of processes for the fruit and vegetable, food, and other production sectors | 50< | Main contractor and supplier SME | 1990 | 18 June 2014 |
| E4 | Manufacturing and electronic services for the fruit and vegetable sector | 20< | Main contractor and supplier SME | 1996 | 19 June 2014 |
| E5 | Palletization systems for the ceramic and fruit and vegetable sectors and all types of industrial packaging | 50< | SME | 1992 | 25 May 2014 |
| E6 | Manufacture of production line machinery for the fruit and vegetable sector | 50< | SME | 1991 | 10 April 2014 |
| E7 | Development and implementation of electronic systems for sizing fruit and vegetables | 5< | SME, start-up | 2012 | 15 April 2014 (two interviews with different persons) |

**Table A1.** *Cont.*

| Enterprise | Activity | Size | DOMC/SME/SU | Date of Creation | Date of the Interview |
|---|---|---|---|---|---|
| E8 | Manufacture of machinery for sorting and packing fresh products | | Main contractor and supplier Large multinational firm | 1966 | 18 June 2014 |
| E9 | Manufacture of machinery for sorting and packaging fruit and vegetables | | Main contractor and supplier Large multinational firm | 1940 | 19 June 2014 |
| E10 | Manufacturing and services for post-harvest products and application machine | 100< | Main contractor and supplier SME | 1956 | 9 April 2006 20 May 2013 1 April 2014 |
| E11 | Manufacturing and services for post-harvest products and application machine | 1000< | Main contractor and supplier Large multinational firm | 1967 | 4 January 2006 18 April 2013 2 April 2014 (two interviews with different persons) |
| E12 | Manufacturing and services for post-harvest products and application machine | 100< | SME | 1964 | 1 March 2011 12 December 2014 |
| E13 | Manufacturing and services for post-harvest products and application machine | 100< | SME Multinational firm | 1980 | 12 April 2006 23 April 2013 16 May 2014 (two interviews with different persons) |

**Table A2.** Characteristics of enterprises and date of the interviews—France.

| Enterprise | Activity | Size | Date of Creation | Date of the Interview |
|---|---|---|---|---|
| F1 | Helicopter manufacturer | 10,000< | 1989 | 28 February 2008 22 September 2010 |
| F2 | Multisector equipment manufacturer | 3000< | 1961 | 20 March 2008 |
| F3 | Multisector equipment manufacturer: machining of prototype parts, safety valves | 250 < purchased by an international group | 1988 | 3 April 2008 2 September 2010 |
| F4 | Multisector equipment manufacturer: robots, naval drones, aviation | 250< | 1936 | 5 March 2008 24 May 2012 |
| F5 | Multisector equipment manufacturer: onboard electronic and electromechanic equipment | 50< | 1947, bankruptcy and repurchased in 1994 by engineers | 12 April 2008 |
| F6 | Multisector equipment manufacturer: hydraulic hoses, new materials | 50< | 1971 (2002 taken over by the son and new CEO) | 16 April 2008 12 December 2014 |
| F7 | Multisector equipment manufacturer: engine testing hardware and software | 20< | 2007 | 16 April 2008 1 March 2011 12 December 2014 |

<div align="center">**Table A2.** *Cont.*</div>

| Enterprise | Activity | Size | Date of Creation | Date of the Interview |
|---|---|---|---|---|
| F8 | Multisector equipment manufacturer: study, machining, prototypes, mechanical parts, tooling | 10< | 1990 | 27 May 2008 |
| F9 | Multisector equipment manufacturer: cockpit dashboard systems | 50< | 2005 | 1 April 2008 9 December 2014 20 July 2017 |
| F10 | CTS, multisector equipment manufacturer: climatic and thermal electronic and IT equipment, sensors | 50< | 1995 | 12 February 2008 |
| F11 | Multisector systems manufacturer: pedagogical manufacturing machines (cosmetics sector), sensors and electronic systems for the nuclear sector and aviation | 100< | 2004 Nuclear and aviation section purchased in 2010 by an SME of the mechanical sector | 17 March 2008 14 December 2014 |
| F12 | Aviation equipment: exhaust systems with silencers | 10< | 1996 | 11 September 2008 5 December 2014 |
| F13 | Ultralight helicopter systems manufacturer | 20< | 2001 | 28 September 2008 |
| F14 | Systems manufacturer, design and manufacture of equipment for paramotors | 10< | 1995 | 3 July 2008 |

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
