# Peer review of "Rebuilding a Cluster While Protecting Knowledge within Low-Medium-Tech Supplier SMEs: A Spanish and French Comparison"

_sustainability, doi:10.3390/su132011313_

Round 1

Reviewer 1 Report

  1. At the introduction section, I recommend a summary of study methods and highlights of contribution/intended contributions.
  2.  why the choice of France and Spain? I can see you have given some descriptions of the technological centres in the method section, it would be ideal to give the rational for selecting these two countries despite the many countries in Europe.
  3. Also, I noticed the tables 2 and 3 have the type of IP protection for only larger firms, is that right? If that’s the case what about small and medium-sized firms?

Reviewer 2 Report

Thank you for this article Rebuilding a cluster while protecting knowledge within low-medium-tech supplier SMEs: a Spanish and French comparison which adds to the literature on IP, innovation and SMEs from a European perspective.

First, I suggest you make it more express in the abstract that you have conducted a multi-case longitudinal study involving a #number of companies.

Good opening sentence to the article. 

I suggest you need to better define Low tech SMEs and Medium Tech SMEs and relate this more directly to your case studies. 

You state, "In our international comparative perspective, the industrial dynamic and survival of territories are at the core of the analyses." [lines 324 325]

Please clarify why you have selected the two clusters and why it is important to compare them.   It seems rather random at first glance.   The  comparative methodology is not entirely clear.  What is the economic background of Valencia and Provence?  

Nor is the rationale for "rebuilding a cluster" clear.   What is being rebuilt and why?  

Comforting the results of this initial literature line 45 – Typo, should be ‘Confirming’ (page 2 of 32) I believe.

This sentence lacks clarity [lines 324-325] – what do you mean by ‘survival of territories’?  

"We have designed the research issue of this latest wave of research in the continuity of the first one with the aim to understand how the owner managers of these firms through interactions over time have organized their business and cooperation in order to protect their knowledge related to technological innovation within an open innovation paradigm." [Lines342-345]

The above is an over long sentence, I suggest you divide into two sentence for clarity of expression.

Section 4 industrial and Societal contexts (Valencia and Provence) could be expanded.  I suggest it would  be helpful to align  more explicitly the discussion with each of the three propositions.

Section 5 You could introduce the content in the Tables 1 and 2 in more detail at the outset.  I found it difficult to discern what the numbers mean in your discussion and the precise data being reported.

Cite the Spanish patent law directly at lines 669-670 And directly compare with French patent law.  Which precise legal provision are you referring to? 

Perhaps introduce a third table in the discussion section to aid / summarise the key themes to emerge from the comparison between the Spanish and French experience.

I also suggest you need more detail regarding precise similarities and differences between Spanish and French approach and strategies by SMEs and the regional dynamics of innovation and the relationship between IP and open innovation.  In my view this is still  a bit vague.

Finally, in the conclusion you state that the most important difference in the behaviours of this type of firm between the two regions, based on the comparison is societal rather than sectoral.   As a reviewer, I don’t have a clear sense of specifically what this means and which precise behaviours are or what driving the similarities and differences.  

Also, are you able to suggest any wider conclusions for EU SMEs or SMEs globally as a result of this research? 

The article has clear potential but I would advocate the above revisions prior to publications in order to make it a stronger piece.  
